# Allele frequency divergence reveals ubiquitous influence of positive selection in *Drosophila*

Jason Bertram[1,2]*

**1** Environmental Resilience Institute, Indiana University, Bloomington, Indiana, United States of America,
**2** Department of Biology, Indiana University, Bloomington, Indiana, United States of America

* jxb@iu.edu

## Abstract

Resolving the role of natural selection is a basic objective of evolutionary biology. It is generally difficult to detect the influence of selection because ubiquitous non-selective stochastic change in allele frequencies (genetic drift) degrades evidence of selection. As a result, selection scans typically only identify genomic regions that have undergone episodes of intense selection. Yet it seems likely such episodes are the exception; the norm is more likely to involve subtle, concurrent selective changes at a large number of loci. We develop a new theoretical approach that uncovers a previously undocumented genome-wide signature of selection in the collective divergence of allele frequencies over time. Applying our approach to temporally resolved allele frequency measurements from laboratory and wild *Drosophila* populations, we quantify the selective contribution to allele frequency divergence and find that selection has substantial effects on much of the genome. We further quantify the magnitude of the total selection coefficient (a measure of the combined effects of direct and linked selection) at a typical polymorphic locus, and find this to be large (of order 1%) even though most mutations are not directly under selection. We find that selective allele frequency divergence is substantially elevated at intermediate allele frequencies, which we argue is most parsimoniously explained by positive—not negative—selection. Thus, in these populations most mutations are far from evolving neutrally in the short term (tens of generations), including mutations with neutral fitness effects, and the result cannot be explained simply as an ongoing purging of deleterious mutations.

## Author summary

Natural selection is the process fundamentally driving evolutionary adaptation; yet the specifics of how natural selection molds the genome are contentious. A prevailing neutralist view holds that the evolution of most mutations is essentially random. Here, we develop new theory that looks past the stochasticity of individual mutations and instead analyzes the behavior of mutations across the genome as a collective. We find that selection has a strong non-random influence on most of the *Drosophila* genome over short

**Data Availability Statement:** All processing and plotting code can be accessed at https://github.com/jasonbertram/polygenic_variance_public/. SNP frequency data were obtained from the open access resources published in the cited studies.

**Funding:** The author(s) received no specific funding for this work.

**Competing interests:** The authors have declared that no competing interests exist.

timescales (tens of generations), including the bulk of mutations that are not themselves directly targeted by selection. We show that this likely involves ongoing positive selection.

## Introduction

One of the central problems of evolutionary biology is to delineate the role of natural selection in shaping genetic variation. Most genetic variation consists of neutral mutations which, though having no appreciable effects on fitness, are not free from the influence of selection. When selection acts on non-neutral mutations, neutral mutations that share similar genetic backgrounds can be dragged along for the ride, a process called linked selection [1]. The extent to which linked selection influences neutral variation is a major point of contention [2, 3]— one with practical implications because putatively neutral mutations are widely used to infer population demographic history [4] and as a baseline for detecting selection [2, 5]. There is also ongoing debate about the particular modes of selection responsible for shaping genetic variation. Negative selection purging the influx of deleterious mutations is probably prevalent [6, 7], but positive selection on rarer advantageous mutations is crucial for adaptive evolution and likely also has a hand in shaping neutral variation [8].

Until recently, the bulk of the evidence entering the above debates rested on patterns of genetic variation measured at single snapshots in time. The interpretation of such evidence is complicated because the prospective signatures of selection are accumulated over an uncertain history during which other confounding processes (e.g. population demography) also shape genetic diversity [5, 9, 10]. Crucially, single snapshot data is unable to reveal what the process of selection is doing at any point in time i.e. selectively changing allele frequencies.

A more direct approach is to analyze allele frequency data gathered from the same population at multiple points in time [10]. Evolve and resequence (E&R) experiments [11–14] and studies on wild populations [15, 16] have identified allele frequency changes associated with rapid phenotypic adaptation. However, determining the full nature and extent of selective allele frequency change has been difficult. Numerous methods exist for inferring selection coefficients from allele frequency time series [10, 17–24], but are only reliable for selection that is strong relative to the intensity of random, non-selective allele frequency change (random genetic drift) and allele frequency measurement error (e.g. due to population sampling or limited sequencing read depth).

This is a major limitation that likely precludes detection of most of the influence of selection. Fitness-relevant traits are often complex (influenced by a large number of genes) and harbor ample genetic variation. Selection on such traits will thus often cause modest allele frequency shifts distributed across many loci rather than be concentrated at a small number of strongly selected loci [25–27]. Moreover, even if some genomic regions harbor strongly selected alleles, much of the associated linked selection could be undetectably weak. Thus, resolving the short-term ($\sim$ tens of generations) influence of selection across the genome remains an important challenge [28].

Here we present a new approach to analyze the genome-wide influence of selection using time-resolved allele frequency data. Our approach capitalizes on a distinctive pattern of among-locus temporal allele frequency divergence that to our knowledge has not previously been described. In contrast with single-locus approaches, this allele frequency divergence is a collective pattern incorporating alleles across the genome. We therefore lose the ability to identify particular loci under selection; in return are able to detect polygenic selective processes that are not detectable with single-locus approaches.

Traditionally the allele frequency variance in a cohort of neutral alleles with initial frequency $p$ is assumed to have the binomial form

$$\text{Var}(\Delta_t p | p) = C_t p(1 - p) \qquad (1)$$

where $\Delta_t p$ denotes the change in allele frequency after $t$ generations, and the variance coefficient $C_t$ is frequency independent [29, Chap. 3]. The allele frequency divergence in Eq (1) is largely a consequence of random genetic drift. However, selection can also cause neutral allele frequencies to diverge. The influential effective population size literature has derived (frequency-independent) expressions for $C_t$ in a wide variety of circumstances [30]. Crucially, a large body of work has attempted to subsume the effects of selection on neutral alleles into the frequency-independent value of $C_t$, including both the effects of unlinked fitness variation [31, 32], and some manifestations of linked selection [6, 33, 34]. The effective population size literature thus views (1) as a broadly applicable model of neutral allele divergence, simply requiring a tuning of $C_t$ to capture the effects of selection on neutral alleles, at least to a first approximation [3, 30].

Here we show, on the contrary, that linked selection causes among-locus neutral allele frequency variance to deviate from the binomial form (1), such that the variance coefficient $C_t$ is frequency-dependent. We use this frequency-dependence to detect the presence of selection, analyze its influence on allele frequencies over time and estimate the typical magnitude of total selection coefficients (capturing both direct and linked selection) across the genome. Applying our approach to E&R and wild *Drosophila* single nucleotide polymorphism (SNP) data we find evidence of strong linked selection affecting most SNPs (although we cannot rule out migratory fluxes in the wild population). We argue that the specific form of frequency-dependence we find implies a substantial role for positive selection.

## Results

### Neutral evolution implies binomial allele frequency variance

The Eq (1) binomial variance classically arises in the neutral Wright-Fisher model, which assumes random sampling of gametes each generation; then $C_t = 1 - (1 - 1/2N)^t$ where $N$ is the (diploid) population size. In its basic form the neutral Wright-Fisher model entails a number of biological simplifications including random mating, constant $N$, non-overlapping generations, and the absence of fitness differences between individuals. Many of these assumptions can be relaxed without affecting the binomial form of Eq (1), at least approximately for large $N$ and over long timescales [30, 35]. Similarly, much of the justification for Wright-Fisher as a biologically valid description of genetic drift is derived from its equivalence to a broader class of drift models in the limit of large $N$ and slow allele frequency change (the diffusion limit [36]). Here we are interested in shorter time scales ($\leq$ tens of generations), and want our approach to be applicable to small laboratory populations ($< 10^3$ individuals). We therefore evaluate the validity of Eq (1) more generally.

An enormous variety of purely neutral genetic drift models have binomial variance [37]. This includes the Cannings model, which represents neutrality using a general exchangeability assumption that allows for arbitrary offspring number distributions [38]. Binomial variance thus accommodates fundamental deviations from Wright-Fisher such as "sweepstakes" reproduction in high-fecundity organisms [39]. However, due to the presence of fitness variation in adapting populations, neutral mutations do not evolve according to "pure drift" of the sort studied in ref. [37], even if unlinked from alleles under selection [31]. In particular, the Cannings model is not applicable because exchangeability precludes fitness variation between individuals.

We show that binomial variance applies quite generally for neutral alleles unlinked from selected loci (A in S1 Text). In short, we use a generalized exchangeability argument to show that binomial variance holds in the presence of fitness variation provided that the neutral alleles under consideration are in linkage equilibrium with alleles under selection. Intuitively, linkage equilibrium ensures that the distribution of genetic backgrounds is exchangeable between alternate neutral alleles, even though individual genetic backgrounds are not exchangeable.

Non-binomial variance (equivalently, frequency-dependent $C_t$) thus signifies a violation of generalized exchangeability. The obvious way for this to occur is for allele frequency change to have a nonzero bias; this could be due to linked selection, migration, mutation bias or gene drive. Additionally, deviations from binomial variance can occur if the population is structured into genetically differentiated demes (B in S1 Text). Below we check for binomial variance empirically and discuss our findings in relation to these factors leading to non-binomial variance, focusing mostly on selection for reasons that will become apparent.

Note that while our exchangeability argument yields Eq (1) with finite variance for finite $N$, in the diffusion limit infinite variance is possible in the Cannings model [37]. None of our results depend on $N \to \infty$ limiting behavior, so we do not discuss this possibility further.

## Selection creates non-binomial allele frequency variance

We now analyze the effects of selection on allele frequency divergence, demonstrating that deviations from binomial variance will often result.

The expected frequency change after one generation due to selection on an allele starting at frequency $p$ is given by

$$E[\Delta_1 p | p, s] = sp(1 - p),$$ (2)

where $s = (\overline{w}_F - \overline{w}_{NF})/\overline{w}$ is the selection coefficient, $\overline{w}_F$ is the mean fitness of the focal allele, $\overline{w}_{NF}$ is the mean fitness of all other alleles at the same locus, and $\overline{w}$ is population mean fitness. Here $s$ is the "total" selection coefficient that captures the net effect of selection at linked loci and the focal locus [40, 41].

Selection generates among-locus divergence of allele frequencies when its strength or sign varies among alleles in a cohort. To quantify this effect, we apply the law of total variance to $\Delta_1 p$ where $s$ is allowed to vary between loci:

$$\mathrm{Var}(\Delta_1 p | p) = E_s[\mathrm{Var}(\Delta_1 p | p, s)] + \mathrm{Var}_s(E[\Delta_1 p, | p, s]).$$ (3)

The second term in Eq (3) represents selective divergence i.e. the deterministic allele frequency divergence created by among-locus variation in $s$. Using Eq (2), it can be written as $\sigma^2(s|p)[p(1 - p)]^2$, where $\sigma^2(s|p)$ is the (possibly frequency-dependent) variance in total selection coefficients among loci with initial frequency $p$. The presence of the $[p(1 - p)]^2$ factor will tend to cause intermediate frequency alleles to have elevated variance relative to the binomial case (Fig 1). Thus, while it is possible for the allele frequency variance created by selection to be binomial, in general it is not. Beyond the tendency for elevated variance at intermediate frequencies, the exact shape and magnitude of the deviation is determined by $\sigma^2(s|p)$.

More generally, allele frequencies are measured $t > 1$ generations apart during which time the selective divergence accumulates. The temporal structure of selection is then important. The total allele frequency change after $t$ generations is the sum over the intervening $t$ generations $\Delta_t p = \sum_{i=0}^{t-1} \delta_i p$, where $\delta_i p = p_{i+1} - p_i$ and $p_i$ is the frequency in generation $i$ ($i = 0, 1, \ldots, t - 1$ counting from the preceding measurement). From Eq (2) we have $\delta_i p = s_i p_i (1 - p_i)$ where $s_i$ is the total selection coefficient in generation $i$. Assuming that the total selection coefficients

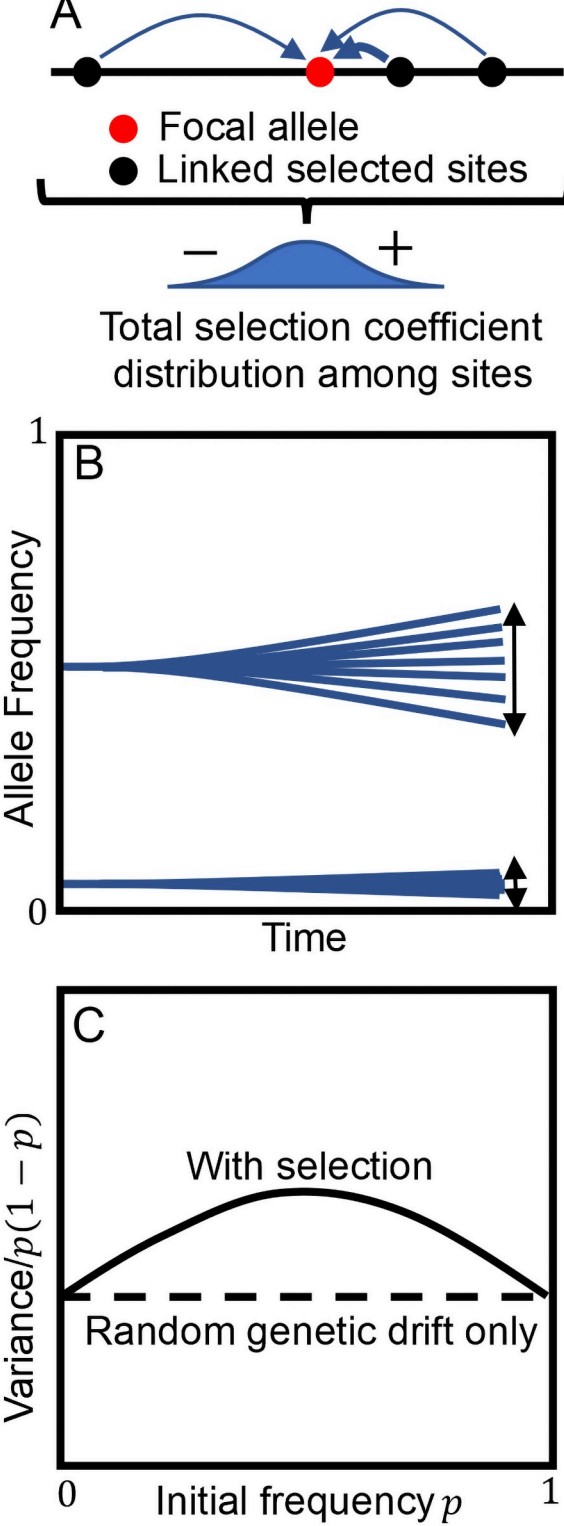

**Fig 1.** (A) The total selection coefficient measures the overall effect of selection on an allele including any associations with other sites under selection. (B) When alleles at different sites have different total selection coefficients, selection generates allele frequency divergence. (C) Compared to the binomial variance (proportional to $p(1-p)$) created by random genetic drift, the selective variance tends to be more elevated at intermediate frequencies (proportional to $[p(1-p)]^2$) because the magnitude of selective allele frequency change is proportional to $p(1-p)$.

and total allele frequency change over $t$ generations are small ($\sum_{i=0}^{t-1} s_i \ll 1$ and $|s_i| \ll 1$), the expected allele frequency change is approximately $E[\Delta_t p|p; s_0, \ldots, s_{t-1}] = \sum_{i=0}^{t-1} s_i p_i (1 - p_i) \approx p(1-p)\sum_{i=0}^{t-1} s_i$ (dropping terms of order $s^2$). The selective divergence is then given by

$$
\begin{aligned}
\mathrm{Var}_s(E[\Delta_t p|p; s_1, \ldots, s_{t-1}]) \quad &= [p(1-p)]^2 \mathrm{Var}_s\left(\sum_{i=0}^{t-1} s_i\right) \\
&= [p(1-p)]^2 \left(\sum_{i=0}^{t-1} \mathrm{Var}(s_i) + \sum_{i \neq j} \mathrm{Cov}(s_i, s_j)\right).
\end{aligned}
\tag{4}
$$

The two sums on the right represent respectively: the divergence contribution from fitness variation within intervening generations; and the divergence contribution from temporal consistency in fitness variation across intervening generations.

Sustained selection manifests as positive among-locus temporal covariances $\mathrm{Cov}(s_i, s_j) > 0$. If total selection coefficients were perfectly constant with time these positive covariances would create rapid selective divergence with the allele frequency variance in Eq (4) growing quadratically over time (because there are $t(t-1)$ covariance terms in Eq (4); for further details see C in S1 Text). However, for neutral alleles (the bulk of segregating variants), the temporal covariance between $s_i$ and $s_j$ is expected to decay exponentially with increasing time separation $|j - i|$ due to recombination. In the two-locus case where the neutral allele is hitchhiking with one selected allele, linkage disequilibrium (and thus covariance) decays at rate $\sim (1 - r)^{|j-i|}$ where $r$ is the recombination rate between the two alleles [1]. The multilocus case similarly involves exponential decay averaged over all linked sites under selection [42]. Nevertheless, even if recombination destroys linkage disequilibrium so rapidly that only concurrent generations $|i - j| = 1$ covary, there are still $t - 1$ such pairs contributing to Eq (4). Thus, among-locus temporal autocovariances $\mathrm{Cov}(s_i, s_j)$ can make a substantial contribution to the overall selective divergence.

Alternatively, even if selection fluctuates in such a way that total selection coefficients are temporally uncorrelated $\mathrm{Cov}(s_i, s_j) = 0$, the within-generation selective divergence can still create non-binomial frequency dependence. The variance resulting from this effect accumulates at a slower linear rate with time (because there are $t$ variance terms in Eq (4); C in S1 Text)—a selective random walk [43]. Selection that changes in a more predictable manner could in principle generate no overall divergence at all—if selection reverses direction concurrently at many loci, negative covariances can be created in Eq (4) shrinking the overall divergence.

In addition to the selective divergence described above, selection has another effect in Eq (3): it perturbs the drift contribution to divergence $E_s[\mathrm{Var}(\Delta_t p|p, s_0, \ldots, s_{t-1})]$. This effect occurs when a mean selective bias in the cohort displaces allele frequencies and thus perturbs the effects of drift (regardless of whether there is among-locus variation in total selection coefficients). We show that the selective perturbation to the drift variance has the form $c(1 - 2p)E_s[\sum_{i=0}^{t-1} s_i]$ where $c$ is a frequency-independent constant of order 1 (D in S1 Text). This result assumes that the cohort does not start close to fixation, and is also insensitive to population dynamic specifics if many generations separate measurements ($t \sim 10$ in the data we analyze). For a generational measurement interval ($t = 1$) this result also holds in canonical models (i.e. Wright-Fisher and Moran), but in general it is possible that the exact form of the selective drift perturbation depends on population specifics. In the following analysis the exact expression for the selective drift perturbation will not be important; we only use the fact that it scales with $E_s[\sum_{i=0}^{t-1} s_i]$, which implies that its effects are negligibly small in the populations of interest here (Methods).

Combining variance contributions we have

$$C_t(p) = D_t + D_t c(1 - 2p)E_s\left[\sum_{i=0}^{t-1} s_i | p\right] + p(1 - p)\sigma^2\left(\sum_{i=0}^{t-1} s_i | p\right), \tag{5}$$

where $D_t$ is the frequency-independent variance coefficient in the absence of selection. The variance coefficient $C_t(p)$ is thus partitioned respectively into a frequency-independent genetic drift component, a frequency-dependent selective drift perturbation, and a frequency-dependent selective divergence.

## Positive excess variance indicates positive selection

The deviation from binomial allele frequency variance described in the previous section depends crucially on the among-locus total selection coefficient variance $\sigma^2(s|p)$. This quantity is challenging to analyze because it is determined by the structure of linkage disequilibrium. We thus performed forward-time population genetic simulations using SLiM [44] to supplement our theoretical results (see Methods for simulation details). For simplicity, we focus on three archetypal scenarios in an unstructured, demographically stable population closed to migration: a continual influx of deleterious mutations, no non-neutral mutations (the control case), and a continual influx of unconditionally beneficial mutations. For short we call the first and last of these "negative selection" and "positive selection" respectively. In all three cases we maintain a steady influx of neutral mutations; these constitute the bulk of segregating mutations and therefore dominate the behavior of $\sigma^2(s|p)$. Intuitively we expect that the frequency dependence of $\sigma^2(s|p)$ could be quite different in the negative versus positive selection scenarios, because unconditionally deleterious mutations strong enough to cause detectable allele frequency divergence rarely reach intermediate frequencies, whereas beneficial mutations routinely do so.

To check for binomial frequency variance, we use allele frequencies from two timepoints $t = 10$ generations apart (chosen for compatibility with the empirical data we consider below) to calculate $C_t = \mathrm{Var}(\Delta_t p | p)/p(1 - p)$ for alleles starting at intermediate $0.5 < p < 0.55$ and high $0.9 < p^* < 0.95$ major allele frequencies. We then calculate the "excess variance" $C_t(p) - C_t(p^*)$. We also calculate total selection coefficients for all segregating mutations to investigate how the selective divergence term $p(1 - p)\sigma^2(\sum_{i=0}^{t-1} s_i | p)$ in Eq (5) behaves as a function of $p$. To make the magnitude of the latter easier to interpret, we show total selection coefficient variance on a per-generation scale $\sigma^2(\bar{s}|p) = \sigma^2(\sum_{i=0}^{t-1} s_i | p)/t^2$ where $\bar{s} = \frac{1}{t}\sum_{i=0}^{t-1} s_i$ is the time-averaged total selection coefficient.

According to the theory in the preceding section, selective divergence tends to create positive excess variance due to the $p(1 - p)$ factor in the last term in Eq (5). Our positive selection simulations confirm this prediction, consistently creating positive excess variance (Fig 2A and 2C). On the other hand, there is no consistent deviation from binomial variance in the negative selection simulations: $\sigma^2(\bar{s}|p)$ increases with major allele frequency so rapidly that the overall selective divergence term $\sigma^2(\bar{s}|p)p(1 - p)$ in Eq (5) is independent of frequency (Fig 2A and 2B). While these simulations are obviously simplified, the concentration of selective divergence at low/high frequencies is a general feature of the purging of new deleterious mutations. Thus, selection does generate elevated variance at intermediate frequencies as predicted theoretically, but not just any form selection: it is important that selection be "positive" in the sense of not only eliminating rare variants.

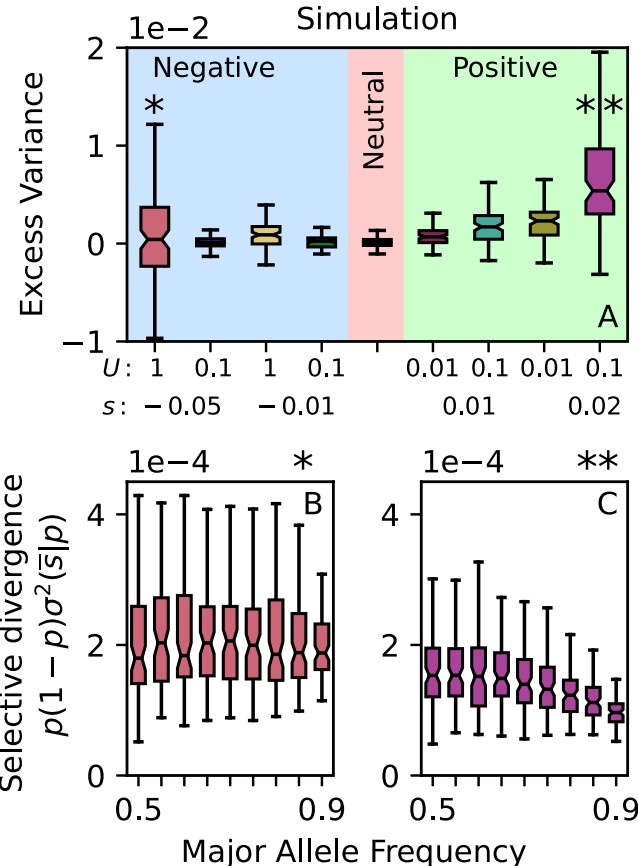

**Fig 2.** (A) Forward-time population genetic simulations consistently show elevated excess variance under positive selection only. Excess variance defined as $C_t(p) - C_t(p^*)$ with major allele frequencies $0.5 < p < 0.55$ and $0.9 < p^* < 0.95$ and $t = 10$ generations. (B) Under strong negative selection (deleterious mutation rate $U = 1$/genome/generation, mutation selection coefficient $s = -0.05$), total selection coefficients are substantial at all frequencies but much stronger for high major allele frequencies resulting in a frequency-independent overall selective divergence $p(1-p)\sigma^2(\bar{s}|p)$ like the neutral case. (C) In contrast, the selective divergence $p(1-p)\sigma^2(\bar{s}|p)$ shows clear frequency dependence under positive selection, thus producing excess variance at intermediate frequencies. Population size $N = 1000$; 100 replicates per parameter combination. Stars indicate which panel A simulations are shown in panels B and C respectively.

## Intermediate frequency alleles have elevated variance in *Drosophila*

We next investigated whether binomial allele frequency variance is observed empirically. In two fruit fly (*D. Simulans*) E&R experiments [11, 12], we observe systematically elevated variance coefficients $C_t$ at intermediate frequencies (Fig 3). We rule out measurement error as driving this pattern, because the major sources of pooled sequencing error (population sampling, read sampling, unequal individual contributions to pooled DNA) also create binomial variance rather than a systematic frequency-dependent bias (E in S1 Text; [45, 46]). We also rule out migration, since these E&R populations are closed. Moreover, as will be discussed in the next section, systematically elevated variance cannot be explained by a few large effect loci, implying that a substantial fraction of SNPs across the genome are involved in the observed pattern. Hence we also rule out mutation bias and gene drive as being the main driver of elevated variance at intermediate frequencies since these processes do not have the requisite scale. Finally, population structure tends to create a variance deficit at intermediate frequencies (B in S1 Text); thus, even if some population structure is present in these closed E&R

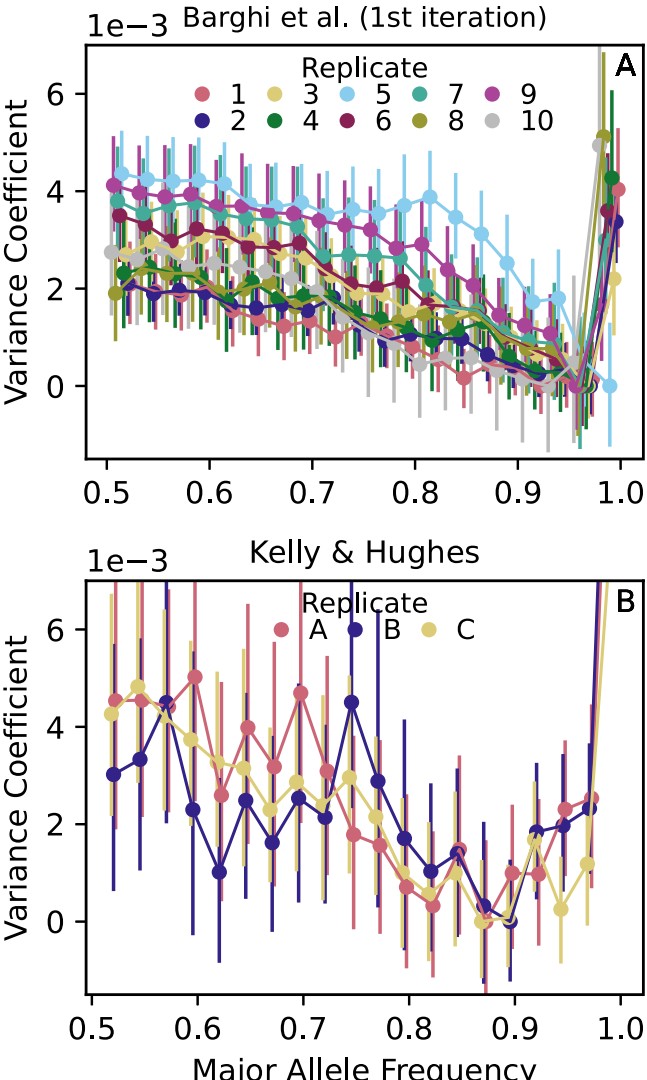

**Fig 3. Intermediate frequency SNPs in E&R *D. Simulans* populations (A [11]; B [12]) have systematically elevated variance coefficients $C_t(p) = \mathrm{Var}(\Delta_t p|p)/p(1-p)$ relative to higher frequency SNPs after one round of evolution and resequencing ($t \approx 10$ in A; $t \approx 15$ in B), inconsistent with the binomial expectation for neutrally evolving alleles (1).** $C_t(p)$ is calculated in 2.5% major allele frequency bins using all SNPs in the genome (circles). Vertical lines show 95% block bootstrap confidence intervals (1Mb blocks). We subtract the constant $\min_p C_t(p)$ from $C_t(p)$ in each replicate to prevent differences in the overall magnitude of $C_t(p)$ between replicates from obscuring $p$ dependence within each replicate.

populations, it would tend to eliminate the observed elevation of variance, not explain it. We deduce that the pattern observed in Fig 3 is due to selection, consistent with the theoretical prediction that selective divergence tends to cause elevated variance at intermediate frequencies.

Similar results are found in a wild *D. Melanogaster* population [15] (S1 Fig), although this population is not closed and elevated variance could also be attributed to migration. The effect of migration on allele frequency divergence can be understood analogously to selection (Eq (3)) as introducing a migration divergence term $\mathrm{Var}(m(p^* - p)|p) = m^2\,\mathrm{Var}(p^* - p|p)$ where $m$ is the proportion of individuals in the focal population replaced by migrants from the source

population each generation, and $p^*$ denotes source population frequencies. The migration divergence thus depends on the structure of differentiation between focal and source populations. The *a priori* expectation is for $\mathrm{Var}(p^* - p|p)$ to be greatest at high $p$ (the opposite of the observed pattern), where the largest differences $p^* - p$ are possible (analogous to the mathematical constraints on $F_{ST}$ [47]). However, since we do not know the structure of population differentiation (or even what the source population might be), we remain agnostic about the influence of migration in the ref. [15] population.

Next we explored the behavior over time of the elevated variance shown in Fig 3 by following its accumulation within a frequency cohort for two studies in which allele frequencies were measured more than twice [11, 15]. Similar to our simulations, at each measured timepoint we quantified the excess variance using the difference $C_t(p) - C_t(p^*)$, where $p$ is the initial frequency of the cohort and $p^* > p$ is a reference frequency. In practice we choose $p = 0.5$ to maximize the contrast with the reference frequency, while $p^* \sim 0.8$–$0.9$ is chosen to be large enough that there is a meaningful contrast with $p = 0.5$ but safely displaced from the $p = 1$ boundary where allele frequency variances are not measured reliably (see sharp increases in Fig 3 as $p \rightarrow 1$).

We find that excess variance accumulates over the course of the entire Barghi et al. [11] E&R experiment (Fig 4A shows one replicate, other replicates are similar; S2 Fig), implying a sustained, polygenic divergence in allele frequencies. This pattern is consistent with the positive $\Delta p$ temporal autocovariances documented in [28]. Sustained divergence is what we expect to occur from selection in a novel but constant laboratory environment.

By contrast, excess variance in wild *D. Melanogaster* populations [15] does not exhibit continual accumulation of excess variance over time, with fluctuations evident in each cohort (Fig 4B). Fluctuations imply a concurrent reversal in the direction of non-neutral allele frequency change across many loci such that non-neutral divergence is partly lost to a subsequent coordinated non-neutral convergence. Bearing in mind that migration may contribute to this pattern, the fluctuations shown in Fig 4 are compatible with temporally fluctuating selection affecting a large proportion of the genome, as proposed by ref. [15]. However, while ref. [15] attributed temporal fluctuations in selection to periodic seasonal change, we do not see a clear annual periodicity in the accumulation of variance. A similar lack of annual periodicity is found in allele frequency temporal autocovariances [28]. These results suggest a more complex selective (or migratory) regime of which seasonal fluctuations are only a part.

## Linked selection strongly perturbs SNP frequencies in *Drosophila*

In the previous section we argued that selection is most likely responsible for elevated allele frequency divergence at intermediate frequencies in three *Drosophila* studies (with the possible exception of the ref. [15] study because of migration). We next used the theory developed above to estimate the typical magnitude of total selection coefficients associated with elevated divergence (we also apply our analysis to ref. [15] supposing that selection was responsible).

We measure the typical intensity of selection using the among-locus standard deviation $\sigma(s|p)$. This quantity determines the selective divergence in Eq (5), and has the convenient property of measuring the absolute magnitude of $s$ regardless of sign. Intuitively, $\sigma(s|p)$ measures the intensity of a collective "polygenic" adaptive response shared across many loci. If a fraction $f$ of loci have $s = 0$, then $\sigma(s|p) = \sqrt{1-f}\,\sigma_{\mathrm{nn}}(s|p)$ where $\sigma_{\mathrm{nn}}(s|p)$ is the standard deviation in $s$ among non-neutral loci. Thus, a substantial fraction of the alleles in a cohort must have nonzero $s$ ($f$ appreciably smaller than 1) for there to be a discernible $\sigma(s|p)$ signal.

We estimate $\sigma(s|p)$ from measured allele frequency divergence using Eq (5). Since we only have measurements separated by $t$ generations, we actually estimate $\sigma(\bar{s}|p) = \sigma(\sum_{i=0}^{t-1} s_i|p)/t$

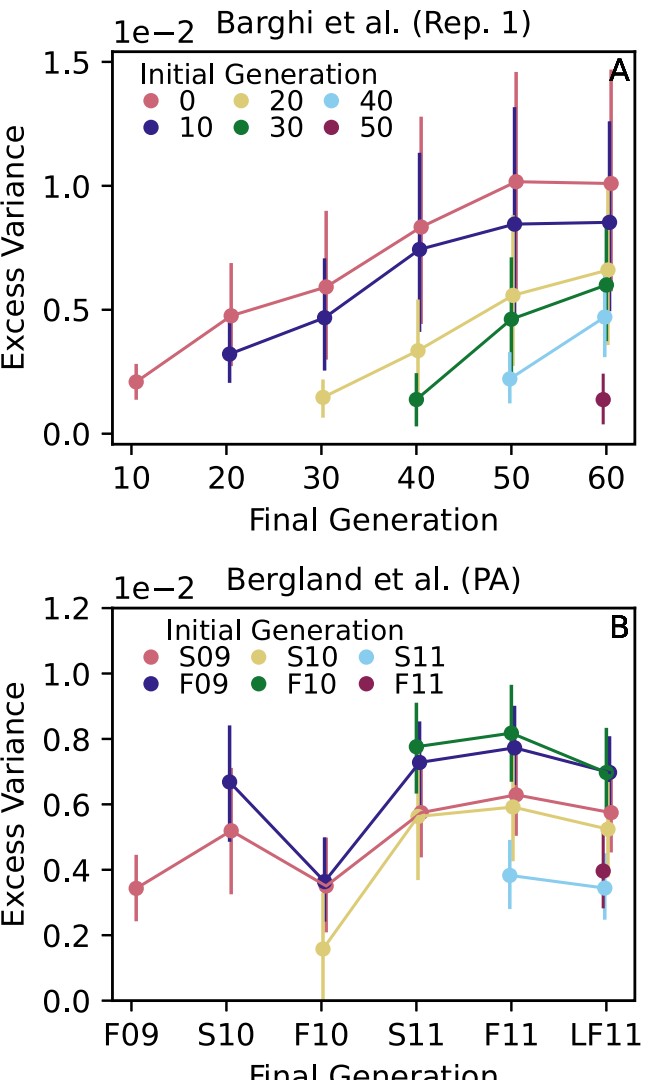

**Fig 4. Excess allele frequency variance (a measure of deviation from neutrality defined as $C_t(p) - C_t(p^*)$) accumulates over time in a *D. Simulans* E&R experiment (A; [11]), but remains relatively flat in a wild *D. Melanogaster* population (B; S = Spring, F = Fall; LF = Late Fall; 09 = 2009 etc.; [15]).** The excess variance is calculated for intermediate frequency alleles falling within a major allele frequency bin at $p = 0.5$. In (A), $p^* = 0.9$ and bin width is 2.5%. In (B), $p^* = 0.8$ and bin width is 5%. Vertical lines show 95% block bootstrap confidence intervals (1Mb blocks).

where $\bar{s}$ is the time-averaged selection coefficient $\bar{s} = \frac{1}{t}\sum_{i=0}^{t-1} s_i$. To estimate $\sigma(\bar{s}|p)$ from Eq (5), we need to eliminate the non-selective divergence contributions of genetic drift $D_t$ and measurement error (which was not included in Eq (5)). In Methods we show that these latter contributions are cancelled out in the excess variance $C_t(p) - C_t(p^*)$, avoiding the complication of independently estimating them. However, some selective divergence is also cancelled out in the difference $C_t(p) - C_t(p^*)$, so that this approach only obtains a lower bound

$$\sigma^2(\bar{s}|p) > \frac{C_t(p) - C_t(p^*)}{t^2 p(1-p)}. \tag{6}$$

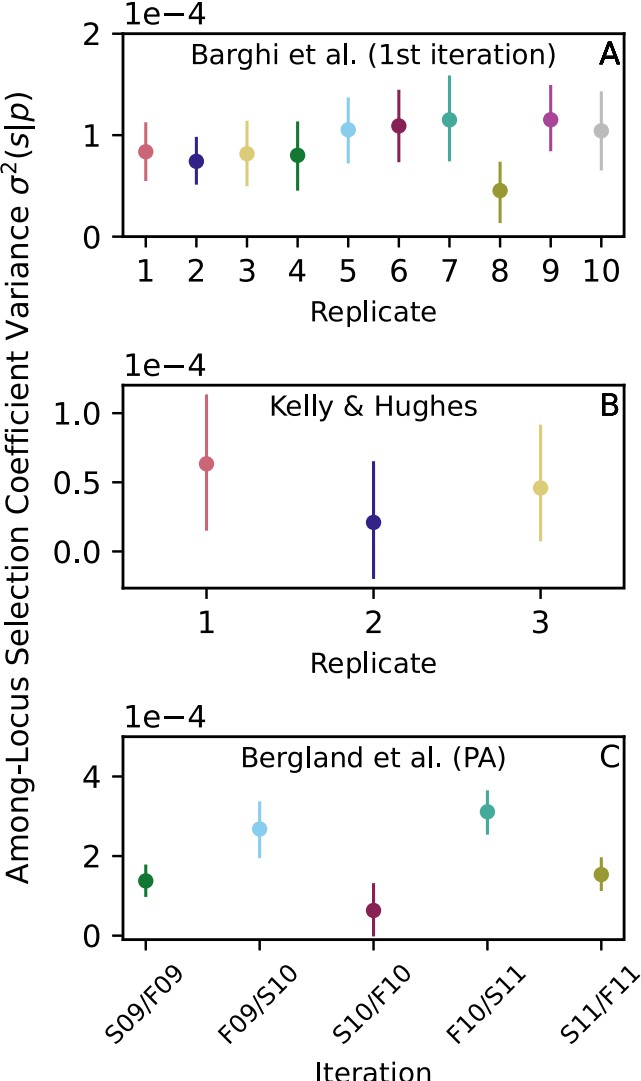

**Fig 5. Total selection coefficients show substantial among-locus variance in *Drosophila*.** (A-C) Lower bound estimates of $\sigma^2(\bar{s}|p)$ calculated from (6) (circles; vertical lines show 95% block bootstrap confidence intervals) are of order $10^{-4}$, which implies typical $s$ values of $\sim 1\%$. Following the original studies [11, 12, 15], we assume $t = 10$ (A); $t = 15$ (B) and $t = 10$ (C; for both summer and winter).

In all three *Drosophila* studies, we find the above lower bound to be of order $10^{-4}$ (Fig 5), implying that total selection coefficients with magnitudes of order $\sigma(\bar{s}|p) \sim 1\%$ are commonplace in the populations considered here.

## Discussion

Several lines of evidence support the view that selection strongly influences genetic variation in *Drosophila* [8, 12, 28, 48]. Our results independently show that even over a short time interval (tens of generations), most intermediate frequency SNPs are influenced by selection—total selection coefficients (which include linked selection) of $|s| \sim 1\%$ are the norm among intermediate frequency SNPs, despite most of these SNPs having no effect on fitness. Since our method relies on contrasting behavior at different frequencies, the effect of selection on extreme

frequency alleles is used as a reference and is therefore not directly inferred. We expect the effects of selection to be even greater at extreme frequencies where most deleterious mutations are segregating and recent neutral mutations are most tightly linked to selected backgrounds.

The power of our approach stems from aggregating allele frequency behavior over many loci, thereby leveraging the sheer number of variants measured with whole-genome sequencing to discern a selective signal. Heuristically, the sampling error in the lower bound estimate (6) is proportional to $1/\sqrt{L}$ where $L$ is the number of independent loci used to estimate $C_t(p)$. With enough sequenced variants ($L \sim 10^5$), selection coefficients of order $|s| \sim 1\%$ should be detectable over a single generation even when allele frequency noise is of comparable magnitude (i.e. read depth and population size $\sim 10^2$; see Methods). Intuitively, variants across the genome experience a detectable non-neutral shift as a collective even though the underlying allele frequency changes may be indistinguishable from drift at individual loci.

Our approach is a departure from the widespread use of frequency-independent $C_t$ for neutral mutations [30]. The variance coefficient $C_t$ can be expressed in terms of the "variance effective population size" $N_e$ as $C_t = 1 - (1 - 1/2N_e)^t$. Thus, selection makes $N_e$ frequency-dependent for neutral mutations over short timescales (i.e. before an appreciable fraction of the alleles in a cohort fix). The origin of this non-binomial allele frequency variance is variation in the selective background of alleles at different loci.

Selection does not need to be consistent over time to have this effect: stochastically fluctuating selection with no temporal consistency can also generate non-binomial allele frequency variance. However, temporally consistent selection generates divergence more rapidly, and temporal covariances can be responsible for most of the selective divergence (Results). Moreover, allele frequency changes $\Delta p$ are correlated over time in the systems analyzed here [28]. Thus, it seems likely that temporally consistent selection is at least partly responsible for the patterns documented here.

Note, however, that in contrast to ref. [28], the temporal covariances relevant to allele frequency divergence in Eq (4) are between total selection coefficients, not $\Delta p$. For among-locus temporal covariances in total selection coefficients to be non-zero it is necessary for those coefficients to vary among loci, whereas $\Delta p$ covariances quantify any temporal consistency in allele frequency change [42]. Thus, $\Delta p$ temporal covariances can theoretically be present without any selective divergence, and vice versa. In practice, the temporal autocovariances in $\Delta p$ must be calculated across three measurement steps e.g. $\text{Cov}(p_t - p_0, p_{2t} - p_t)$. These cross-measurement covariances do not contribute to the divergence observed at $t$ generations, and are only a subset of the covariances contributing to the divergence observed at $2t$ generations (Eq (4)). Therefore, the patterns of variance accumulation documented here are related but not equivalent to the patterns documented in ref. [28]. Temporal autocovariances in $\Delta p$ predominantly capture the extent to which the genome-wide influence of selection has a temporally enduring pattern across measurements. Allele frequency divergence captures the cumulative genome-wide influence of both temporally stable and fluctuating selection between two measurements. The relative contribution from temporal covariances in total selection coefficients depends on the intensity of selective fluctuations as well as the persistence time of linkage disequilibrium (Results), and would require generational allele frequency measurements to quantify.

We found that the frequency structure of allele frequency divergence is informative about the underlying structure of direct selection (Fig 2). Elevated divergence of intermediate frequency alleles is difficult to explain if only negative selection on unconditionally deleterious mutations is occurring. Although selection against an influx of deleterious mutations can generate transient sweep-like behavior for neutral mutations that originate on genetic backgrounds with above-average fitness, this scenario still entails overwhelmingly more influence

on allele frequency dynamics at low/high frequencies compared to intermediate frequencies [49]. More broadly, it may be possible to make more detailed inferences about the structure of direct selection by moving beyond allele frequency variances and analyzing the entire distribution of allele frequency change $\Delta_t p$.

Quantifying the bounds on how much selection is possible, and how much selection actually occurs in natural popoulations, is a long running controversy [50, 51]. The strong total selection coefficients ($|s| \sim 1\%$) we find must predominantly reflect linked selection on neutral SNPs. This implies a substantial risk of overestimating the amount of direct selection when, as is commonly done, selection coefficients are inferred at individual loci and then attributed to direct selection. This "excess significance" is a well known difficulty in E&R experiments [12, 52], and similar challenges have arisen in wild populations [15]. Our results indicate that improving the sensitivity of single-locus selection coefficient inferences, or better controlling for multiple comparisons, will likely not resolve this issue. Our total selection coefficient estimates are also substantially larger than direct selection coefficients of individual alleles estimated from diversity patterns in *Drosophila* [8]. This is consistent with a linkage-centered view of neutral mutation evolution in which the selective background of most neutral mutations contains multiple alleles under selection such that allele frequency behavior is governed by the fitness variation within local "linkage blocks" [53] or larger haplotypes [11].

## Methods

### Simulations

We used SLiM [44] to simulate a closed population with $N = 10^3$ individuals, a 100Mb diploid genome, a recombination rate of $10^{-8}$/base pair/generation, and a neutral mutation rate of $10^{-8}$/base pair/generation. Non-neutral mutations were introduced at rate $U$/chromosome/generation, where in each simulation non-neutral mutations were assumed to have the same fixed selection coefficient. Four background selection regimes ($U = 1, 0.1 \times s = -0.05, -0.01$), one neutral regime ($U = 0$), and four positive selection regimes ($U = 0.1, 0.01 \times s = 0.01, 0.02$) were evaluated (Fig 2). In each regime, 100 replicates were simulated with complete genotypes recorded at generations $10^4$ and $10^4 + 10$, mimicking the $t = 10$ generation interval in the empirical studies after a burn in period of $10N = 10^4$ generations. Total selection coefficients in Fig 2B and 2C computed using Eq (2) from genotype data at generation $10^4$.

### Data processing

SNP frequency data were obtained from the open access resources published in [15] (wild *D. Melanogaster*, 1 replicate, $\sim 5 \times 10^5$ SNPs, 7 timepoints), [11] (*D. Simulans* E&R, 10 replicates, $\sim 5 \times 10^6$ SNPs, 7 timepoints) and [12] (*D. Simulans* E&R, 3 replicates, $\sim 3 \times 10^5$ SNPs, 2 timepoints). We performed no additional SNP filtering. For the [15] data, only SNPs tagged as "used" were included.

### Block bootstrap confidence intervals

We use bootstrapping to estimate the variability of the quantities plotted in Figs 3–5. These quantities are calculated as an average over loci, where nearby loci are unlikely to be statistically independent due to linkage. To account for the non-independence of individual loci when bootstrap sampling, 95% confidence intervals are calculated using a block bootstrap procedure [28]. Each chromosome is partitioned into 1 megabase windows ($\sim 120$ total windows). Bootstrap sampling is then applied to these windows. The plotted vertical lines span the 2.5% and 97.5% block bootstrap percentiles.

## Estimation of the selection coefficient variance

To derive Eq (6) we show that the reference value $C_t(p^*)$ satisfies the inequality

$$
\begin{aligned}
C_t(p^*) \quad &= M + D_t + D_t(1 - 2p^*)E_s\Big[\sum_{i=0}^{t-1} s_i | p^*\Big] \\
&+ p^*(1 - p^*)\sigma^2\Big(\sum_{i=0}^{t-1} s_i | p^*\Big) \\
&> M + D_t
\end{aligned}
\tag{7}
$$

The first line above is Eq (5) evaluated at the reference frequency $p^*$ with an additional measurement error term $M$ included. $M$ is frequency-independent because measurement error is binomial (E in S1 Text; [45, 46]). Eq (7) implies that the reference value $C_t(p^*)$ is an upper bound on the drift and measurement components of $C_t(p)$ for all $p$. Taking the difference $C_t(p) - C(p^*)$, we then have

$$
C_t(p) - C_t(p^*) < p(1 - p)\sigma^2(st|p),
\tag{8}
$$

eliminating $D_t$ and $M$.

To derive Eq (7) we first drop the selective drift perturbation $D_t(1 - 2p^*)E_s[\sum_{i=0}^{t-1} s_i | p^*]$ because it is negligibly small compared to the selective divergence in the populations considered here: $C_t$ (and therefore $D_t$) is of order $10^{-2}$, $E[s_i|p]$ is at most of order $10^{-2}$, and $t \sim 10$; hence $D_t(1 - 2p^*)E_s[\sum_{i=0}^{t-1} s_i | p^*] \sim 10^{-3}$. By comparison, $t^2\sigma^2(\bar{s}|p)$ is of order $10^{-2}$. Second, we have $p^*(1 - p^*)\sigma^2(st|p^*) > 0$; subtracting this term gives the inequality.

## Estimation limits

Our analysis relies on detecting differences in $C_t(p)$ between cohorts with different values of $p$. The ability to detect such differences is determined by the sampling error in $C_t(p)$ arising due to the calculation of $\mathrm{Var}(\Delta_t p|p)$ from a finite number of loci. To estimate this sampling error, we assume that $\Delta_t p$ is approximately normally distributed, in which case the sample variance in $\mathrm{Var}(\Delta_t p|p)$ is $2\mathrm{Var}(\Delta_t p|p)^2/(L - 1) \approx 2\mathrm{Var}(\Delta_t p|p)^2/L$ where $L \gg 1$ is the number of independent loci used to estimate $\mathrm{Var}(\Delta_t p|p)$. The standard error in $C_t(p) = \mathrm{Var}(\Delta_t p|p)/p(1 - p)$ is thus given by $\sqrt{2/L}C_t(p)$. This defines the scale of statistically detectable differences in $C_t(p) - C_t(p^*)$, which in turn determines the statistically detectable lower bound estimate on $\sigma^2(s|p)$ (6). For example, to detect $\sigma^2(s|p) \sim 10^{-4}$ at $p = 0.5$ (i.e. a typical selection coefficient of $\sigma(s|p) \sim 1\%$) after one generation of evolution with $C_1 \sim 10^{-2}$ (i.e. a population sample of $\sim 100$ individuals, an average read depth of $\sim 100$ and fairly strong genetic drift $D_1 \sim 10^{-2}$), we need at least $L \sim 10^5$ independent SNPs.

## Supporting information

**S1 Text. Supplemental text.** This file contains supplemental text sections A-E.
(PDF)

**S1 Fig. Frequency dependence of *C*.** Same as Fig 3 but for the Bergland et al. data. Each curve represents a different seasonal iterate e.g. summer 2009 to fall 2009.
(TIF)

**S2 Fig. All Barghi et al. replicates.** Same as Fig 4A but including all 10 replicates from Barghi et al.
(TIF)

## Acknowledgments

We thank Matthew Hahn for insightful discussions.

## Author Contributions

**Conceptualization:** Jason Bertram.

**Formal analysis:** Jason Bertram.

**Methodology:** Jason Bertram.

**Writing – original draft:** Jason Bertram.

**Writing – review & editing:** Jason Bertram.

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
