## [Decision Letter · Decision Letter 0]

7 Sep 2021

Dear Dr Bertram,

Thank you very much for submitting your Research Article entitled 'Allele frequency divergence reveals ubiquitous influence of positive selection in Drosophila' to PLOS Genetics.

The manuscript was fully evaluated at the editorial level and by independent peer reviewers. The reviewers appreciated the attention to an important problem, but raised some substantial concerns about the current manuscript. Based on the reviews, we will not be able to accept this version of the manuscript, but we would be willing to review a much-revised version. We cannot, of course, promise publication at that time.

If you decide to revise the manuscript for further consideration at PLOS Genetics, please aim to resubmit within the next 60 days, unless it will take extra time to address the concerns of the reviewers, in which case we would appreciate an expected resubmission date by email to plosgenetics@plos.org.

[LINK]

We are sorry that we cannot be more positive about your manuscript at this stage. Please do not hesitate to contact us if you have any concerns or questions.

Yours sincerely,

Alex Buerkle

Associate Editor

PLOS Genetics

Bret Payseur

Section Editor: Evolution

PLOS Genetics

This manuscript has been reviewed by two referees, both of whom find the theoretical, methodological, and empirical results to be interesting. The reviews offer a number of suggestions for improvement of the manuscript, including encouragement to move material from the Supplement/Appendix into the main paper and to develop one or more visualizations to communicate the core result. They also raise some questions that are important for clarity and evaluation of the manuscript.

Reviewer's Responses to Questions

**Comments to the Authors:**

Reviewer #1: Bertram develops a theoretical partitioning of allele frequency change to distinguish genetic drift from natural selection from time series data, which here is SNP allele frequencies scored over time within an evolving population. The signal is that selection elevates the magnitude of change at intermediate allele frequencies relative to drift. He applies the approach to allele frequency measurements from laboratory and wild Drosophila populations and finds that selection has genome-wide effects. This paper is worthy of publication because it illustrates an important way that linked selection / genetic draft cannot be subsumed into an “effective population size effect.” The applications to Drosophila demonstrate the genomic extent of this process. I have two major suggestions and several minor comments.

I. The paper has the structure of a short-format paper (e.g. Nature or PNAS). Plos Genetics allows a more rigorous treatment of a topic within the main body of the paper. For this reason, I think Appendix S3 should be moved into the main paper. The equations of S3 provide the reader with a far more explicit understanding of the parts of eq 4. It is important to see the sources of variability (and covariation) in selection coefficients, both among loci and through time, as determinants of outcomes. It also sets up an important contrast for the application to Drosophila data. Temporally fluctuating selection (of a specific flavor) is implicated for the Bergland et al field studies but not for the laboratory Evolve-and-Resequence studies. These differences are discussed (e.g. lines 207 onward, lines 307-325) but could be explained more clearly by reference to previous in-text equations.

II. The semantic classifications of selection are inconsistent and confusing. For example, Lines 12-14 set up selection as positive or negative. Directional selection is always positive on one allele and negative on the other. Here, the idea seems to involve the initial allele frequency of the favored allele. If so, that should be defined explicitly. At other points in the manuscript, the author contrasts positive selection to “purifying selection.” The latter term is typically associated with mutation-selection balance. Purifying selection is where selection acts to eliminate variation. Its usually considered as the alternative to balancing selection (where selection acts to maintain polymorphism), but that may be the meaning here.

Perhaps for terminology reasons, I also struggled to follow the section of Lines 253-272. The point seems to be that “selection acting continuously to purge deleterious alleles” does not explain the results. Are the hypothesized mutations unconditionally deleterious? If so, they would never have made it into the studies in the first place. Mutations rare at time zero and then stay rare would be filtered during variant calling and thus never make it into the analyzed datasets. If the mutations were at intermediate frequency at the initiation of the study (time = 0), they would not likely have been unconditionally deleterious. One allele may have become deleterious in the lab environment in which the population was maintained if it was different (or even just more constant) than the ancestral environment. However, I am not sure how this scenario is distinct from “positive selection.”

Other comments:

Lines 27-29: “Numerous methods exist for inferring selection coefficients from allele frequency time series [10, 17–24], but are only reliable for selection that is strong relative to the intensity of random, non-selective allele frequency change (random genetic drift).”

For samples from natural populations (millions of individuals), the noise from finite sampling (limited numbers of individuals taken plus limited sequencing coverage) is likely to be more important problem than genetic drift in terms of the power of outlier tests.

Line 370: Does “eliminate” mean ignore here? I cannot tell if this is a mathematical operation or the term has simply been dropped.

Reviewer #2: This MS develops a framework for testing temporal fluctuations in allele frequency due to selection. This framework uses the the variance in allele frequencies, conditional on starting frequency and shows using theory, simulation, and real data that strong temporally variable selection occurs over short time periods. Overall the MS is well written and I think that the results are relevant to a broad audience.

I think that the MS could be restructured a bit to make it more approachable to the non-theoretician. I still struggle to understand why the excess variance is frequency dependent. Can the author put together a conceptual figure that will help motivate an intuitive understanding of the results? I think that this would be crucial for this paper to be accessible to the PloS G readership.

I also think that the paper could be re-ordered. First theory, then simulations, then data. Having the simulations at the end feels like an afterthought. However for people who might struggle with the maths, the simulation results very striking and might help the reader trust the importance of the results better.

Minor comments

Line 143: "that is not important to our results". Do you mean "that is important to our results"? If you mean what you wrote, why should I care to read something that is not important?

Line 185-186: "Similar results are found in a wild D. melanogaster population (not shown in Fig 1)." It is a little funny to read this sentence and then not see the results. Why didn't you include this dataset in Fig 1?

Fig 4. What the 1e-2, 1e-4, and stars refer to?

**Have all data underlying the figures and results presented in the manuscript been provided?**

Reviewer #1: None

Reviewer #2: Yes

PLOS authors have the option to publish the peer review history of their article (what does this mean?). If published, this will include your full peer review and any attached files.

Reviewer #1: No

Reviewer #2: No

---

## [Editor Report · Decision Letter 1]

22 Sep 2021

Dear Dr Bertram,

We are pleased to inform you that your manuscript entitled "Allele frequency divergence reveals ubiquitous influence of positive selection in Drosophila" has been editorially accepted for publication in PLOS Genetics. Congratulations!

Yours sincerely,

Alex Buerkle

Associate Editor

PLOS Genetics

Bret Payseur

Section Editor: Evolution

PLOS Genetics

Comments from the reviewers (if applicable):

I appreciate the author's clear and direct responses to the suggestions for the improvement of the manuscript. I have reviewed these and believe they fully address all of the points that arose in the reviews.

**Data Deposition**

http://datadryad.org/submit?journalID=pgenetics&manu=PGENETICS-D-21-00936R1

**Press Queries**

---

## [Editor Report · Acceptance letter]

27 Sep 2021

PGENETICS-D-21-00936R1 

Allele frequency divergence reveals ubiquitous influence of positive selection in Drosophila 

Dear Dr Bertram, 

We are pleased to inform you that your manuscript entitled "Allele frequency divergence reveals ubiquitous influence of positive selection in Drosophila" has been formally accepted for publication in PLOS Genetics! Your manuscript is now with our production department and you will be notified of the publication date in due course.

With kind regards,

Anita Estes

PLOS Genetics

On behalf of:
